# OpenReview forum: "RMB: Comprehensively benchmarking reward models in LLM alignment"
_ICLR.cc/2025/Conference — ICLR 2025 Poster_

### Official Review · Reviewer_qTwD · 2024-10-27

**Soundness:** 3
**Presentation:** 4
**Contribution:** 3
**Rating:** 6
**Confidence:** 4

**Summary:**

The paper proposes RMB, a comprehensive benchmark for evaluating reward models (RMs) in LLM alignment. The dataset contains comprehensive real-world scenarios on helpfulness and harmfulness and responses generated by different LLMs. The paper validates the benchmark's effectiveness by showing a correlation with downstream alignment performance.

**Strengths:**

- The paper is well-structured and clearly written, explaining the methodology and results effectively.
- RMB effectively addresses the limitations of developing reward models that align with the objective, whether helpfulness or harmlessness. Correlation evaluations show that the reward model's performance on RMB can reflect the performance of the downstream aligned model more accurately. This would enable researchers to evaluate and iterate on reward models more efficiently before training a more computationally challenging model alignment.
- The paper provides solid empirical evidence supporting the effectiveness of RMB. The experiments on pairwise and best-of-N accuracy cover different generative and discriminative reward models and provide a good insight into their usefulness as judges.

**Weaknesses:**

- The reliance on LLM-generated response in the dataset curation may pose potential long-term limitations. As reward models are often used to align newer LLMs, using responses from current LLMs to evaluate them could create a circular dependency. In other words, this benchmark may not be able to differentiate reward models that could guide LLMs that are beyond current capabilities. In the next iterations, it would be nice to incorporate human-generated responses to ensure more diverse sources of responses.
- The correlation evaluation is leveraging best-of-N as an alignment, which is not the only way people are using the reward models. Would it be possible to gather a set of reward models and aligned model pairs that leverage the reward model during training and see the correlation there?
- nit: Mu et al.. citation is missing a year (L229)

**Questions:**

See Weakness.

---

> ### Author Response · Authors · 2024-11-21
> **Response to Reviewer qTwD (1/2)**
>
> Thank you sincerely for your thoughtful and positive feedback on our work. Below, we have provided a detailed explanation for your concern as follows. In accordance with your suggestion, we have made revisions to our manuscripts. We added the potential **limited diversity of LLM responses** to the **Limitation Section** of our work and **fixed the missing year of the citation** as you kindly pointed out. All revisions made to the paper are highlighted in blue for your ease of reference. Please do not hesitate to let us know if you have any further questions.
>
> **P1: Enhancing diversity on response distribution considering the fast development on LLM capabilities (Response to Weakness 1)**
> >Thank you for pointing out this important consideration. Your feedback is highly valuable and provides meaningful direction for the continued development and improvement of our benchmark.
> >
> >We do share the same awareness of the importance of ensuring a broad distribution of model responses during the dataset curation process. To promote **diversity** in response distributions, we **included LLMs across different capability gradients** in our dataset.
> Additionally, we included responses from **state-of-the-art (SOTA) models** (e.g., GPT-4o and Claude-3.5), considering the ongoing improvement of base models in the community. This inclusion helps **delay the long-term issue** you raised.
> >
> >We promise to update the dataset in future iterations by incorporating responses from newer SOTA models and human responses to maintain its diversity and utility. Besides, thanks to the automation of data construction processes, the cost of such updates is not particularly high.
> >
> >**Action taken**: We added this concern to the **Limitation section** and highlighted it in blue.

---

> ### Author Response · Authors · 2024-11-21
> **Response to Reviewer qTwD (2/2)**
>
> **P2: Response to the limitation in RLHF experiments (Response to Weakness 2):**
> > - **Reasons to use BoN as alignment**:
> >Considering resource constraints and the unstable nature of full RL tasks, we did not conduct a complete RL task to validate the correlation between RMB and downstream tasks, which is one of the limitations of our work. We chose Best-of-N (BoN) as an alignment approach for the following reasons:
> >   - **Established use in efficient model alignment**: Inference-time BoN has been widely used in a series of popular works [1,2,3] and has been shown to outperform RL in WebGPT.
> >   - **Theoretical equivalence to RL solutions**: BoN has been proven to be asymptotically equivalent to KL-constrained RL solutions [4], and the BoN policy is nearly optimal in terms of win rate versus KL divergence [5].
> >   - **Simplicity and robustness**: BoN has a simpler parameter setup compared to other RL algorithms, eliminating the uncertainty introduced by parameter tuning and its potential influence on the results. Instead, other RL algorithms, such as PPO, are influenced by numerous factors beyond the RM itself, which complicates the accurate verification of the reward model’s specific role in the process.
> >
> > - **Additional experiment conducted:**
> To better validate the correlation between RMB and downstream tasks and further address the reviewer's concern, **we conducted an additional alignment experiment using rejection sampling fine-tuning (RFT) during the rebuttal phase**. RFT aligns models by updating the parameters of policy models with the optimal samples selected by the RM during training, a method adopted in popular works such as Llama 2 [6] and RAFT [7].
> >
> >   The experimental procedure is as follows:
> >  1. We sampled prompts from TriviaQA, GSM8k, MATH, BBH, DROP, and AGIEval.
> >  2. Using these prompts, we conducted RFT on two policy models ,ChatGLM-3-6B and Yi-1.5-6B-Chat ( the same policy model we used in Section 5). Specifically, we fine-tuned the policy models on the RM-selected best-of-N samples for one iteration.
> >  3. We evaluated the aligned models on MixEval-hard, calculated their rankings, and measured the correlation between RM rankings and aligned models rankings (as described in Section 5).
> >
> >   The table below shows the correlation between RMB ranking results and downstream task performance when RFT is used as the RM for alignment tasks, in comparison with Reward Bench. (We only conducted comparison on helpfulness subset due to the limited computing resources) We can see RMB positively correlates the task and outperform the previous work.
> >
> >   | Corr.                  | Yi-1.5-6B-Chat Bo5 | Yi-1.5-6B-Chat Bo20 | ChatGLM3-6B Bo20 |
> >   |------------------------|---------------------|---------------------|------------------|
> >   | RMB-helpfulness        | 0.384              | 0.487              | 0.453           |
> >   | RewardBench-chathard   | 0.238              | 0.131              | 0.361           |
> >
> >   Due to time and resource limitations during the rebuttal period, we regret that we are unable to include additional RLHF experiments (e.g., using PPO) at this stage. Such experiments are computationally intensive and inherently unstable. However, we fully acknowledge that this is an important direction for future research, and we plan to explore it further. Despite this limitation, we firmly believe that our work serves as a valuable preliminary exploration of the relationship between reward model evaluation and downstream RL tasks, providing meaningful contributions to the research community.
> >
>
> **P3: Addressing the missing year in citation (Response to Weakness 3):**
> >Thank you very much for your careful review. We have added the missing publication year in the revised manuscript.
>
> ---
> [1] Nakano, Reiichiro, et al. "Webgpt: Browser-assisted question-answering with human feedback." arXiv preprint arXiv:2112.09332 (2021).
>
> [2] Bai, Yuntao, et al. "Training a helpful and harmless assistant with reinforcement learning from human feedback." arXiv preprint arXiv:2204.05862 (2022).
>
> [3] Zhang, Lunjun, et al. "Generative verifiers: Reward modeling as next-token prediction." arXiv preprint arXiv:2408.15240 (2024).
>
> [4] Yang, Joy Qiping, et al. "Asymptotics of language model alignment." arXiv preprint arXiv:2404.01730 (2024).
>
> [5] Gui, Lin, Cristina Gârbacea, and Victor Veitch. "BoNBoN Alignment for Large Language Models and the Sweetness of Best-of-n Sampling." arXiv preprint arXiv:2406.00832 (2024).
>
> [6] Touvron, Hugo, et al. "Llama 2: Open foundation and fine-tuned chat models." arXiv preprint arXiv:2307.09288 (2023).
>
> [7] Dong, Hanze, et al. "Raft: Reward ranked finetuning for generative foundation model alignment." arXiv preprint arXiv:2304.06767 (2023).

---

> > ### Comment · Reviewer_qTwD · 2024-11-26
> >
> > Thank you the authors for a detailed response and additional changes. My questions have been addressed.

---

> > > ### Author Response · Authors · 2024-11-27
> > > **Thank you for your response**
> > >
> > > Thank you for your valuable feedback! We’re pleased to hear that your questions have been addressed and hope the revisions meet your expectations. Should you have any additional suggestions, we’re always open to further discussion :-)

---

### Official Review · Reviewer_Nun4 · 2024-11-03

**Soundness:** 3
**Presentation:** 2
**Contribution:** 3
**Rating:** 6
**Confidence:** 4

**Summary:**

The paper proposes RMB (Reward Model Benchmark) which is a benchmark for evaluation of reward models used in LLM alignment. It consists of 49 real–world scenarios between the two proposed categories “helpfulness” and “harmlessness”. They use two methods of evaluation: pairwise comparison and best-of-N evaluation, which tests the models’ ability to choose the best response from multiple candidates. Another key finding is that generative models show strong performance, often outperforming discriminative models. Further, they show positive correlation between benchmark performance and downstream alignment tasks.

**Strengths:**

RMB presents many strengths, especially against its main predecessor, RewardBench. Primarily, the dataset size is much larger than any similar datasets used to evaluate reward models. The paper does thorough investigation of best-of-N evaluation, which related papers had proposed as a future direction of research. The authors do a good job of categorization and subcategorization, to ensure broad coverage across different tasks. The benchmark shows good progress towards a useful proxy for downstream model performance. Further, the paper is clearly written and has good formatting.

**Weaknesses:**

There are a couple of weaknesses in the paper. The human annotator sample size is small relative to the size of the dataset. The correlation analysis focuses mainly on the best-of-N sampling rather than RLHF, which is acknowledged in the paper as a limitation. While mentioning that it is hard for a model to be both competitive in judging helpfulness and harmlessness, the authors don’t deeply explore the trade-offs between the two metrics. The authors used various models in the categorization and filtering steps, but the scoring was only done by GPT4, which might introduce bias. There are also a few styling errors and typos throughout the paper that would need to be cleaned up prior to approval.

**Questions:**

- Were any alternatives to GPT-4 scoring considered? How might the benchmark differ if using other models (or ensembles) or human annotators? (I understand human annotation is costly)

- Did you observe any patterns when best-of-N evaluation diverged significantly from pairwise evaluation?

- What recommendations would you make for effectively using RMB during the reward model development process? Is it simply meant to be used as a validation set?

- What tradeoffs between helpfulness and harmlessness exist? Why do you think they seem to be mutually exclusive?

- Do you see any negative impacts or possible misuses of the benchmark?

---

> ### Author Response · Authors · 2024-11-21
> **Response to Reviewer Nun4 (1/6)**
>
> We sincerely thank the reviewer for providing valuable feedback. We detailed our response below point by point. Necessary revisions to our manuscript has been made and been highlighted in blue for your convenience. In **Section 3.3** , we added the clarification on the **human annotation**. **The diverged pattern of BoN** and **the explanation about the tradeoff between helpfulness and harmlessness** has been added in **Section 4.2** and **Appendix D** as a deeper explanation of our key findings. We have carefully re-reviewed our manuscript and corrected all **styling errors** and **typos**. Please kindly let us know whether you have any further concerns :).
>
> **P1: A clarification on the human annotation sample size (Response to weakness 1)**
> > We recognize that 200 human-annotated samples might appear limited compared to the scale of RMB. To clarify, these samples were specifically sampled to validate the agreement between GPT4 annotation and human annotations under the final RMB data distribution. Given the significant time and cost constraints of human annotation, we ensured these samples were highly representative to maximize their impact.
> >
> >During the course of our work, aiming **to iteratively validate and improve the annotation quality**,  we also utilized a **separate hold-out set of human annotations**, which is distinct from the 200 human-annotated samples mentioned in the manuscript that were used to evaluate the final dataset’s annotation quality,:
> > - **Improving inter-annotator agreement Using a held-out dev set**:  Before initiating large-scale GPT-4 annotations, we curated a task pool consisting of 32 prompts for harmlessness and 47 prompts for helpfulness, forming a total of **600 prompt-response pairs**. Three annotators with undergraduate degrees from different fields were tasked with scoring the helpfulness and harmlessness of the responses to these prompts. To guide the scoring process, we employed the following approach: in addition to key features provided by GPT-4, we included five additional key features generated by Claude-3-opus. Annotators were instructed to review the key features, select the five they considered most relevant to the query, and then make a final decision on the response’s rating. We compared the annotators’ Spearman correlation coefficients to evaluate agreement, achieving an inter-annotator correlation of **70% for helpfulness evaluation** and **86% for harmlessness**.
> > - **Optimizing and validating the AIF algorithm quality using a golden set**: We developed a golden set based on the aforementioned human annotations. This golden set was used to iteratively optimize the AIF algorithm by comparing its performance against human annotations, thereby improving the annotation quality.
> >
> >   - **Golden set construction**: The golden set was constructed from instances where all three annotators consistently agreed on the preferred choice in each pair comparison from the dev set. This process resulted in **471 triplets** of (prompt, chosen, rejected) for helpfulness and **411 triplets** for harmlessness.
> >   - **Evaluation and optimization**: We compared the following approaches on the golden set:
> >       - Two-stage scenario-based **GPT-4 judging** (as described in the main text),
> >       - Vanilla pointwise **GPT-4 judging** (single-stage, no scenario-based scoring, based on manual rules),
> >       - Two-stage scenario-based **Claude-3-opus judging**.
> >
> >   The correlation of these annotation methods with human annotators on the golden set is shown in the table below. Through **iterative optimization** of the AIF algorithm on the golden set, we ultimately selected the two-stage GPT-4 judging approach as the annotation method.
> >
> >    | Method | Agreement on Helpfulness | Agreement on Harmlessness |
> >   |-|-|--|
> >   | GPT4 with human     | 84.71% | 97.57%  |
> >   | Single stage with human| 76.01% | 85.64%  |
> >   | Claude with human   | 85.35% | 87.10%  |
> >   | GPT4 with Claude    | 75.80% | 86.62%  |
> >
> >**Action taken**: The held-out dev set and golden set mentioned above were both derived from human annotations and are **distinct from the 200 human-annotated samples discussed in the main text**. Due to space constraints, the explanation about them was initially included in the **Appendix B.4**. In the revised version, we have emphasized the importance of this part by integrating it into **Section 3.3** the main text. The improvements have been highlighted accordingly.

---

> ### Author Response · Authors · 2024-11-21
> **Response to Reviewer Nun4 (2/6)**
>
> **P2: Response to the limitation in RLHF experiments (Response to Weakness 2):**
> > - **Reasons to use BoN as alignment**:
> >Considering resource constraints and the unstable nature of full RL tasks, we did not conduct a complete RL task to validate the correlation between RMB and downstream tasks, which is one of the limitations of our work. We chose Best-of-N (BoN) as an alignment approach for the following reasons:
> >   - **Established use in efficient model alignment**: Inference-time BoN has been widely used in a series of popular works [1,2,3] and has been shown to outperform RL in WebGPT.
> >   - **Theoretical equivalence to RL solutions**: BoN has been proven to be asymptotically equivalent to KL-constrained RL solutions [4], and the BoN policy is nearly optimal in terms of win rate versus KL divergence [5].
> >   - **Simplicity and robustness**: BoN has a simpler parameter setup compared to other RL algorithms, eliminating the uncertainty introduced by parameter tuning and its potential influence on the results. Instead, other RL algorithms, such as PPO, are influenced by numerous factors beyond the RM itself, which complicates the accurate verification of the reward model’s specific role in the process.
> >
> > - **Additional experiment conducted:**
> To better validate the correlation between RMB and downstream tasks and further address the reviewer's concern, **we conducted an additional alignment experiment using rejection sampling fine-tuning (RFT) during the rebuttal phase**. RFT aligns models by updating the parameters of policy models with the optimal samples selected by the RM during training, a method adopted in popular works such as Llama 2 [6] and RAFT [7].
> >
> >   The experimental procedure is as follows:
> >  1. We sampled prompts from TriviaQA, GSM8k, MATH, BBH, DROP, and AGIEval.
> >  2. Using these prompts, we conducted RFT on two policy models ,ChatGLM-3-6B and Yi-1.5-6B-Chat ( the same policy model we used in Section 5). Specifically, we fine-tuned the policy models on the RM-selected best-of-N samples for one iteration.
> >  3. We evaluated the aligned models on MixEval-hard, calculated their rankings, and measured the correlation between RM rankings and aligned models rankings (as described in Section 5).
> >
> >   The table below shows the correlation between RMB ranking results and downstream task performance when RFT is used as the RM for alignment tasks, in comparison with Reward Bench. (We only conducted comparison on helpfulness subset due to the limited computing resources) We can see RMB positively correlates the task and outperform the previous work.
> >
> >   | Corr.                  | Yi-1.5-6B-Chat Bo5 | Yi-1.5-6B-Chat Bo20 | ChatGLM3-6B Bo20 |
> >   |------------------------|---------------------|---------------------|------------------|
> >   | RMB-helpfulness        | 0.384              | 0.487              | 0.453           |
> >   | RewardBench-chathard   | 0.238              | 0.131              | 0.361           |
> >
> > Due to time and resource limitations during the rebuttal period, we regret that we are unable to include additional RLHF experiments (e.g., using PPO) at this stage. Such experiments are computationally intensive and inherently unstable. However, we fully acknowledge that this is an important direction for future research, and we plan to explore it further. Despite this limitation, we firmly believe that our work serves as a valuable preliminary exploration of the relationship between reward model evaluation and downstream RL tasks, providing meaningful contributions to the research community.
> >
> > ---
> >[1] Nakano, Reiichiro, et al. "Webgpt: Browser-assisted question-answering with human feedback." arXiv preprint arXiv:2112.09332 (2021).
> >
> >[2] Bai, Yuntao, et al. "Training a helpful and harmless assistant with reinforcement learning from human feedback." arXiv preprint arXiv:2204.05862 (2022).
> >
> >[3] Zhang, Lunjun, et al. "Generative verifiers: Reward modeling as next-token prediction." arXiv preprint arXiv:2408.15240 (2024).
> >
> >[4] Yang, Joy Qiping, et al. "Asymptotics of language model alignment." arXiv preprint arXiv:2404.01730 (2024).
> >
> >[5] Gui, Lin, Cristina Gârbacea, and Victor Veitch. "BoNBoN Alignment for Large Language Models and the Sweetness of Best-of-n Sampling." arXiv preprint arXiv:2406.00832 (2024).
> >
> >[6] Touvron, Hugo, et al. "Llama 2: Open foundation and fine-tuned chat models." arXiv preprint arXiv:2307.09288 (2023).
> >
> >[7] Dong, Hanze, et al. "Raft: Reward ranked finetuning for generative foundation model alignment." arXiv preprint arXiv:2304.06767 (2023).

---

> ### Author Response · Authors · 2024-11-21
> **Response to Reviewer Nun4 (3/6)**
>
> **P3: A deeper explanation of the trade-off between helpfulness and harmlessness (Response to weakness 3)**
> >**From a phenomenological perspective**, we analyzed the top ten RMs based on their overall rankings on the RMB benchmark and calculated the **Spearman correlation between their individual rankings across the two dimensions**, as Table blow shows. The resulting correlation coefficient was **-0.57**, indicating a negative correlation. This suggests that, among RMs with strong overall capabilities, those that perform well in helpfulness tend to rank lower in harmlessness, and vice versa.
> > | RM| Eurus-RM-7b | Starling-RM-34B | internlm2-7b-reward | Llama3.1-70b-Instruct | Mistral-Large-2407 | Qwen2-72B-Instruct | GPT4o-2024-0513 | Claude-3.5-sonnet | skyword-critic-llama3.1-8B | skyword-critic-llama3.1-70B |
> > |-|---|--|--|---|--|--|--|--|---|--|
> > | Ranking on Helpfulness| 2 | 9 | 8  | 4 | 3 | 5  | 7      | 1| 10     | 6|
> > | Ranking on Harmlessness | 9 | 2 | 7  | 8 | 5 | 3  | 1      | 10      | 6      | 4|
> >
> > **From a theoretical perspective**, The underlying reasons for the trade-off between helpfulness and harmlessness can be follows:
> > - **Intrinsic conflict between the two objectives**: The underlying reason for this challenge stems from an intrinsic conflict between the two objectives[1,2]. For example, if a model refuses to answer an unsafe question from a user, such as “How to build a bomb,” it meets the harmlessness requirement but fails to be helpful. Helpfulness aims for the model to respond to and fulfill human requests, whereas harmlessness requires the model to identify unsafe user intentions and, in certain cases, deny the request. Balancing these two goals often results in a trade-off, such as models overly refusing requests to enhance safety [3].
> > - **Pitfalls of reward hacking**: prior research on reward hacking has shown that models may overfit superficial features, such as response length [4,5]. Comparing the stylistic preferences of the two objectives reveals that helpfulness favors detailed answers, whereas harmlessness emphasizes rejection. We hypothesize that without targeted multi-objective training or strong generalization, reward models may struggle to learn these differing styles effectively.
> >
> > Existing research often categorizes sub-goals under a single reward objective (e.g., helpfulness) to improve interpretability, such as correctness and verbosity [6], but **developing a reward model that captures diverse and conflicting human values** remains a critical challenge. **We took the conflict into account when designing our benchmark** and provide a prior tool to evaluate a reward model’s ability to balance it. Unlike previous datasets with binary safety standards, we ensured harmlessness annotations did not compromise helpfulness. For instance, between equally harmless responses, one offering more guidance scores higher—e.g., a privacy-related query receives higher scores for appropriate guidance over simple refusal (Appendix I.1.2).
> >
> >**Action taken**: We have revised the manuscript to include this comparison in Section 4.2 and Appendix D. We hope the analysis could offer deeper insights and contribute more meaningfully to the community.
> >
> >---
> >[1] Bai, Yuntao, et al. "Training a helpful and harmless assistant with reinforcement learning from human feedback." arXiv preprint arXiv:2204.05862 (2022).
> >
> >[2] Ganguli, Deep, et al. "Red teaming language models to reduce harms: Methods, scaling behaviors, and lessons learned." arXiv preprint arXiv:2209.07858 (2022).
> >
> >[3] Touvron, Hugo, et al. "Llama 2: Open foundation and fine-tuned chat models." arXiv preprint arXiv:2307.09288 (2023).
> >
> >[4] Dubois, Yann, et al. "Length-controlled alpacaeval: A simple way to debias automatic evaluators." arXiv preprint arXiv:2404.04475 (2024).
> >
> >[5] Eisenstein, Jacob, et al. "Helping or herding? reward model ensembles mitigate but do not eliminate reward hacking." arXiv preprint arXiv:2312.09244 (2023).
> >
> >[6] Wang, Haoxiang, et al. "Interpretable Preferences via Multi-Objective Reward Modeling and Mixture-of-Experts." arXiv preprint arXiv:2406.12845 (2024).

---

> ### Author Response · Authors · 2024-11-21
> **Response to Reviewer Nun4 (4/6)**
>
> **P4: Addressing concerns about potential bias from using GPT-4 for scoring ( Response to Weakness 4)**
> >We acknowledge that relying solely on GPT-4 for annotations may introduce some biases. We have taken the following measures to mitigate biases introduced by GPT-4 and have demonstrated that GPT-4 exhibits a high level of consistency with human annotations.
> >
> >**Use of a human-annotated golden set**: As described earlier in P1, we employed a human-annotated golden set to iteratively optimize the AIF algorithm, improving annotation quality and reducing bias. The two-stage GPT-4 scoring method we finally applied for labeling achieved high consistency with human annotations on the golden set.
> >
> >**Majority voting with alternative models**: To further validate our results, we tested the RMB pairs on two large models, LLaMA3-70B-Instruct and Qwen2-72B-Instruct, using majority voting. Specifically, each model performed **10 pairwise comparisons for every RMB pair**, and the majority voting results were taken as the annotations from these models. The consistency between the majority voting results and GPT-4 annotations is shown in the table. The agreement rates, both between individual models’ majority votes and GPT-4 annotations, as well as across all three annotation methods, demonstrate the reliability of the annotation results.
> > ||Agreement on Helpfulness | Agreement on Harmlessness |
> > |-|-|-|
> > | GPT4 with llama majority | 87.13% | 88.86%  |
> > | GPT4 with qwen majority | 88.41% | 88.72%  |
> > | GPT4, llama majority and qwen majority | 82.19% | 84.97% |
> >
> >Additionally, an intuitive concern might be that GPT-4 judging could cause GPT-4 to rank disproportionately high as a generative reward model on the leaderboard. However, our tests revealed that Claude achieved the highest score in helpfulness, while GPT-4 ranked only 7th. This indicates that our carefully curated dataset cannot be fully exploited by GPT-4, indirectly demonstrating that GPT-4’s biases have been effectively suppressed.
> >
> >Thanks for your valuable comment, recognizing and mitigating the potential bias in AI feedback is a fundamental part of scalable LLM development. We also regard this as our future research direction.
>
> **P5: Cleaning up the styling errors and typos ( Response to Weakness 5)**
> >Thans for your valuable time and detailed feedback. We thoroughly reviewed our manuscripts again and cleaned up the styling errors and typos in Figure 2, Section 3.1 and Section 7. The revision has been highlighted. Besides, we normalize the case in the titles.
>
> **P6: Discussion on the alternatives to GPT4 scoring (Response to Q1)**
> >We explored various annotation strategies to improve quality, with the results reaffirming the reliability of GPT-4 scoring:
> >
> > - **Comparison with alternative annotation methods**: As discussed in responses to P1 and P4, we considered two alternatives: Claude-3-opus annotations and majority voting based on LLaMA3-70B-Instruct and Qwen2-72B-Instruct. The tables in P1 and P4 shows the consistency comparisons. Claude-3-opus annotations show a similar level of agreement with human annotations as GPT-4, and the majority voting results from the LLaMA and Qwen models also exhibit high agreement with GPT-4 annotations across all pairs in RMB.
> >
> > - **Model-based annotation aggregation as alternative**: We explored a model-based annotation aggregation approach, using **the consistency of majority voting results as a measure of GPT-4 annotation confidence**. This confidence was then incorporated into the calculation of evaluation metrics. We found that this modification **does not significantly affect the benchmark rankings**, as the following table shows the overall score and the ranking difference. **Figure 24 in Appendix E** shows the **downstream task correlations** averaged across all alignment experiment settings for both the original GPT-4-annotated rankings and the confidence-incorporated rankings, indicating no significant divergence, **with the original performing even better**. The **detailed metrics and results** are provided in Appendix E in our manuscript.
> >| | 0| 1 | 2| 3| 4 | 5 | 6 | 7| 8 | 9 | 10 | 11| 12 | 13  | 14 | 15 | 16 | 17| 18  |
> >|:---|:--|:--|:-|:--|:-|:--|:--|:-|:--|:--|:-|:--|:-|:--|:--|:--|:-|:--|:-|
> >| RMs in new rank| Claude-3-5-sonnet | Qwen2-72B-Instruct | GPT-4o-2024-05-13 | Starling-RM-34B | Mistral-Large-2407 | Llama3.1-70B-Instruct | skywork-reward-Llama3.1-8B | skyword-critic-llama3.1-70B | Eurus-RM-7b | internlm2-7b-reward | skywork-critic-llama3.1-8B | ArmoRM-Llama3-8B-v0.1 | internlm2-20b-reward | gemini-1.5-pro | skywork-reward-Gemma-2-27B | Mixtral-8x7B-Instruct-v0.1 | Llama3.1-8B-Instruct | tulu-v2.5-13b-preference-mix-rm | llama2-70b-chat |
> >| Overall Score| 0.78 | 0.76| 0.75| 0.75 | 0.73| 0.73 | 0.72 | 0.72  | 0.71 | 0.7| 0.69| 0.67 | 0.65| 0.64 | 0.63 | 0.62 | 0.51| 0.46 | 0.46  |
> >| Rank Diff  | +3 | - | -2 | -1 | 0 | +1| -1 | +2| -1| -1 | +2| -1| -1  | +2 | -1| -1   | -| - | -|

---

> ### Author Response · Authors · 2024-11-21
> **Response to Reviewer Nun4 (5/6)**
>
> **P7: Observed patterns when best-of-N evaluation diverged significantly from pairwise evaluation (Response to Q2)**
> > - **The key and most important pattern is that BoN rankings exhibit a stronger correlation with downstream task performance compared to pairwise evaluations**, as concluded in Section 5.2.
> >
> > - From the test results, we identified several specific patterns:
> >
> >   - **BoN scores are lower than pairwise scores, indicating higher test difficulty, especially for generative RMs:**
> >     For each reward model, the scores on the BoN test set are consistently lower than on the pairwise test set. This is intuitive, as a BoN list involves multiple pair comparisons (each winner must outperform multiple losers). We calculate the average differences between BoN and pairwise test scores for discriminative and generative reward models across the two alignment objectives (helpfulness and harmlessness) in the table below:
> >
> >     | | Discriminative RM | Generative RM |
> >     |-|--|--|
> >     | Helpfulness     | 0.167   | 0.191  |
> >     | Harmlessness    | 0.158   | 0.222  |
> >
> >     We find that generative RMs experience more significant score reductions than discriminative RMs, especially in harmlessness evaluation. This phenomenon is validated across various scenarios as **Figure 21 in our manuscript** shows, which compares the differences in BoN test results relative to pairwise evaluation for the two types of reward models in fine-grained task scenarios across the two alignment objectives.
> >
> >   - **Higher differentiation in BoN compared to pairwise:**
> >     The **standard deviation** of model scores on the BoN test set is consistently higher than on the pairwise test set for both helpfulness and harmlessness, as Table 3 shows. This suggests that BoN is a more discriminative evaluation method, as it provides greater differentiation among models.
> >
> >     | | Helpful-BoN | Helpful-pairwise | Harmlessness-BoN | Harmlessness-pairwise |
> >     |-|---|---|----|---|
> >     | Standard Deviation | 0.121| 0.076| 0.132| 0.069|
> >
> > **Action taken:** The higher difficulty and greater differentiation of BoN compared to pairwise evaluations highlight the potential of this testing method. We greatly appreciate your question. We have rewritten the part "Comparison between Pairwise and BoN" in **Section 4.2** (at the end of Page 7) to include the discussion. Due to space limitation, tables, figures, and detailed analysis are included in **Appendix D**.
>
> **P8: Recommendations for effectively using RMB during the reward model development process (Response to Q3)**
>
> > Thank you for your thoughtful question about RMB's role in reward model development. RMB is not merely a validation set; validation sets are typically in-domain and used for adjust methods and model parameters to find a configuration that suits the current training data while avoiding overfitting to it.
> >
> > In contrast, RMB evaluates a reward model’s ability to generalize to **diverse and unseen data**, which is critical in Offline RL. RMs often assess out-of-distribution data during RL training without real-time updates. Testing this capability ensures robustness and effectiveness during its utilization.
> >
> > We recommend using RMB in the following scenarios:
> > - **Algorithmic iterative optimization:** Refine reward models to ensure robust performance across tasks.
> > - **Ablation studies:** Conduct ablation experiments to understand the impact of different components of an RM. For example, investigate whether multi-objective optimization algorithms help balance the trade-offs between helpfulness and harmlessness, or whether robustness optimization enhances performance across various conditions.
> > - **Pre-alignment evaluation:** Most importantly, and our original motivation for developing RMB, is its use for pre-alignment evaluation. As a benchmark highly correlated with downstream task performance, RMB helps assess the reward model before alignment training. Given that alignment training involves numerous variables and is resource-intensive, evaluating the reward model beforehand can help isolate and reduce uncertainties related to parameter optimization and resource consumption.

---

> ### Author Response · Authors · 2024-11-21
> **Response to Reviewer Nun4 (6/6)**
>
> **P9: Explanation to the trade-off between helpfulness and harmlessness (Response to Q4)**
> >According to our understanding, this question is highly related to the discussion in **Weakness 3**. Please refer to **P3** for our response to this issue.
> >If you have any additional questions, please feel free to let us know.
>
> **P10: Potential negative impact or possible misuse of RMB (Response to Q5)**
> >**Over-optimization and data contamination:** One potential misuse is over-optimizing models specifically for scenarios covered by RMB. If developers tailor their reward models too closely to excel on the benchmark, the models may lose generalization ability and underperform on unseen or evolving real-world tasks. To prevent this, RMB should be used as a diagnostic tool rather than the sole basis for model optimization.
> >
> >**Data obsolescence:** While we have included state-of-the-art generative models, such as GPT-4o, to cover a broad data distribution and evaluate reward models’ ability to distinguish between varying response quality, model capabilities will continue to evolve. This may lead to data distribution shifts outside RMB’s coverage. We plan to regularly update our benchmark to address potential data obsolescence as models advance.
> >
> >**Potential bias:** Human values are diverse, and while we have validated the consistency between model annotations and human preferences and used context-based AIF methods to provide AI judges with general HH principles, bias may still persist. Addressing the diversity of human values is a significant challenge in reward model development and will be a focus of our future work.

---

> ### Author Response · Authors · 2024-11-28
> **Still happy to have further discussion**
>
> Dear Reviewer Nun4,
>
> We hope this message finds you well! If this email reaches you during your holiday or outside your usual working hours, please accept our apologies for the interruption.
>
> We just want to kindly follow up to ensure that we’ve addressed any remaining concerns or questions you might have. We are still here and welcome any further discussion or feedback, as your insights are incredibly valuable to us.
>
> Thank you sincerely for all the time and effort during the review process.
>
>
> Best,
>
> Authors of Submission 11004

---

> ### Author Response · Authors · 2024-12-02
> **A Friendly Reminder: Discussion Window Ending**
>
> Dear Reviewer Nun4,
>
> I hope this message finds you well.
>
> As the Author-Reviewer Discussion phase is set to conclude in 24 hours (December 2nd AoE), we would like to kindly follow up again to ask if our responses have adequately addressed your concerns. If there are any outstanding points or further clarifications required, we would be happy to discuss them promptly.
>
> Thank you once again for your time and valuable feedback throughout this process.
>
> Best,
>
> Authors of Submission 11004

---

### Official Review · Reviewer_N8sV · 2024-11-04

**Soundness:** 3
**Presentation:** 3
**Contribution:** 3
**Rating:** 6
**Confidence:** 4

**Summary:**

The paper presents RMB, a benchmark evaluating reward models (RMs) across 49 real-world scenarios, improving alignment assessments for LLMs. RMB reveals generalization issues in current RMs and explores factors affecting generative RM effectiveness.

**Strengths:**

1. The paper is written well and is easy to understand.
2. The studied problem is significant..
3. The results seem to outperform the SOTA datasets of the reward model evaluation.

**Weaknesses:**

1. The paper currently includes some discussion related to benchmark comparisons in Section 5.2, particularly with RewardBench. However, a more explicit comparison of the features and approaches of existing benchmarks early in the paper would better highlight the novelty of this work. Rather than relying on experimental results to convey superior performance, detailing how our model’s capabilities differ from those of previous benchmarks would strengthen the paper's contribution.
2.  In the conclusions about evaluation results, statements like “It is hard for an RM to be both competitive in judging helpfulness and harmlessness” and “Generative models show great promise in reward modeling” could be enhanced by including an explanation of the underlying reasons for these findings. Adding these insights may help the community engage more deeply with the work, offering valuable perspectives that could inspire further exploration beyond the results alone.

**Questions:**

Please refer to the weakness, I'm happy to modify my rate based on the response of the authors and refer to other reviewers' comments.

---

> ### Author Response · Authors · 2024-11-21
> **Response to Reviewer N8sV (1/3)**
>
> We sincerely thank you for providing thoughtful and constructive feedback. Please kindly find our response to your comments below. We have included the **comparison between RMB and previous benchmarks** in **Table 1 referred by Section 1**. The deeper analysis of our key findings has been included in **Section 4.2** and **Appendix C**. All revisions made to the paper are highlighted in blue for your ease of reference. We hope that our response satisfactorily addresses the issues you raised. Please feel free to let us know if you have any additional concerns or questions :-)
>
> **P1: A comparison of features and methods with previous benchmarks. (Response to Weakness 1)**
>
> >We sincerely appreciate the reviewer’s insightful suggestion. Our initial emphasis on experimental results aimed to highlight RMB’s practical impact and effectiveness. However, we now realize that an early comparison of features would better showcase why RMB performs so effectively. Indeed, our comprehensive task coverage, advanced evaluation paradigms, and nuanced harmlessness testing are key factors driving this success.
> >
> >We compared RMB with the only existing benchmark, RewardBench [1], as well as 3  prevalance preference datasets, namely, Ultrafeedback [2], Helpsteer2 [3], and HH-RLHF [4] in terms of features and methods. The advantages of RMB can be summarized as follows, and Table blow provides a comprehensive and clear comparison (**Table 1 at the top of Page 3** provides a more intuitive and visually clear comparison; we kindly invite you to refer to it ):
> > - **Comprehensive data distribution**: RMB covers 49 fine-grained, real-world scenarios, significantly expanding the diversity of tasks compared to existing datasets. This allows for a more comprehensive test of reward model generalization.
> > - **Advanced evaluation paradigms**: RMB introduces both Pairwise and Best-of-N (BoN) evaluations. BoN evaluation, which correlates better than the pairwise evaluation with downstream tasks, is not available in prior work.
> > - **Nuanced harmlessness testing**: Unlike previous works that only provide binary harmlessness labels or simply lack this dimension, RMB includes 12 detailed harmlessness scenarios to assess nuanced safety failures.  This characteristic helps us identify the RM’s **generalization deficiencies in harmlessness** evaluation across different scenarios.
> >
> > | Datasets/Benchmark | OOD tests | Task Granularity | Prompt Sources       | Evaluation Paradigm | Harmlessness Testing            |
> > |---------------------|-----------|------------------|----------------------|---------------------|----------------------------------|
> > | Ultrafeedback       | No        | Coarse-grained   | Synthetic + Real     | Pairwise            | Not included                    |
> > | Helpsteer2          | No        | No division      | Real-world           | Pairwise            | Not included                    |
> > | HH-RLHF             | No        | No division      | Real-world           | Pairwise            | Simple binary labels            |
> > | RewardBench         | Yes       | Coarse-grained   | Synthetic + Real     | Pairwise            | Simple binary labels            |
> > | RMB                 | Yes       | Fine-grained     | Real-world           | Pairwise & BoN      | Detailed, nuanced evaluation    |
> >
> > **Action taken**: We have revised the manuscript to include this comparison in **Section 1** ( **Table 1 on the top of Page 3**). Thanks for your constructive consideration. This enhancement will better emphasize the novelty and breadth of RMB.

---

> ### Author Response · Authors · 2024-11-21
> **Response to Reviewer N8sV (2/3)**
>
> **P2: A deeper explanation of our key findings (Response to Weakness 2)**
> > Thank you for your kind suggestion. We will analyze our key findings point by point to provide deeper insights and contribute more meaningfully to the community.
> >
> > **Analysis of  for the trade-off between helpfulness and harmlessness**
> > - **An additional phenomenological analysis of the helpfulness-harmlessness trade-off**
> > We analyzed the top ten reward models (RMs) based on their overall rankings on the RMB benchmark and calculated the **Spearman correlation between their individual rankings across the two dimensions**, as Table blow shows. The resulting Spearman correlation coefficient was **-0.57**, indicating a negative correlation. This suggests that, among reward models with strong overall capabilities, those that perform well in helpfulness tend to rank lower in harmlessness, and vice versa.
> > | RM| Eurus-RM-7b | Starling-RM-34B | internlm2-7b-reward | Llama3.1-70b-Instruct | Mistral-Large-2407 | Qwen2-72B | GPT4o-2024-0513 | Claude-3.5-sonnet | skyword-critic-llama3.1-8B | skyword-critic-llama3.1-70B |
> > |--|--|--|--|---|--|--|--|--|---|--|
> > | Ranking on Helpfulness| 2 | 9   | 8  | 4 | 3 | 5  | 7  | 1| 10  | 6|
> > | Ranking on Harmlessness  | 9 | 2  | 7  | 8 | 5 | 3  | 1  | 10  | 6  | 4|
> >
> > - **Underlying reasons**:
> >   - **Intrinsic conflict between the two objectives**: The underlying reason for this challenge stems from an intrinsic conflict between the two objectives [4, 5]. For example, if a model refuses to answer an unsafe question from a user, such as “How to build a bomb,” it meets the harmlessness requirement but fails to be helpful. Helpfulness aims for the model to respond to and fulfill human requests, whereas harmlessness requires the model to identify unsafe user intentions and, in certain cases, deny the request. Balancing these two goals often results in a trade-off, such as models overly refusing requests to enhance safety [6].
> >   - **Pitfalls of reward hacking**: Prior research on reward hacking has shown that models may overfit superficial features, such as response length [7, 8]. Comparing the stylistic preferences of the two objectives reveals that helpfulness favors detailed answers, whereas harmlessness emphasizes rejection. We hypothesize that without targeted multi-objective training or strong generalization, reward models may struggle to learn these differing styles effectively.
> > - **Future direction**:
> > Existing research categorizes sub-goals under a single reward objective (e.g., helpfulness) to improve interpretability, such as correctness and verbosity [9], but overlooks conflicts between objectives. **Developing a reward model that captures diverse and conflicting human values** remains a critical challenge. **We took the conflict into account when designing our benchmark** and provide a prior tool to evaluate a reward model’s ability to balance it. Unlike previous datasets with binary safety standards, we ensured harmlessness annotations did not compromise helpfulness. For instance, between equally harmless responses, one offering more guidance scores higher—e.g., a privacy-related query receives higher scores for appropriate guidance over simple refusal (Appendix I.1.2).

---

> ### Author Response · Authors · 2024-11-21
> **Response to Reviewer N8sV (3/3)**
>
> > **Analysis of the promising performance on generative RM**
> > - **Underlying reasons**:
> >   - **Harnessing core generative abilities**: Generative RMs bring their foundational capabilities—such as instruction-following and chain-of-thought (CoT) reasoning—into the reward modeling process. These abilities enable the models to interpret complex prompts and generate contextually rich responses, which is not possible for discriminative RMs [10].
> >   - **Explicit and interpretable reasoning**: Generative RMs often require detailed evaluation criteria and follow specified instructions to execute a series of reasoning steps. This externalized reasoning process is more rubric-driven and interpretable compared to discriminative models, where all reasoning must be performed implicitly within a single forward pass [11].
> >   - **More aligned with how humans make preference judges**: We human beings always conduct a reasoning process to make preference judges (e.g., think about the pros and cons). The above characteristics are well aligned with this process.
> > - **Future direction**:
> > Our findings highlight the value of **leveraging generative models’ language capabilities for preference judgment**. Prior work shows that integrating generative abilities, such as natural language critiques or next-token prediction, can improve reward models [10, 11]. This raises deeper questions about **key capabilities for reward modeling**: for instance, could enhancing LLMs’ reasoning abilities further improve preference modeling?
> >
> -----
> [1] Lambert, Nathan, et al. "Rewardbench: Evaluating reward models for language modeling." arXiv preprint arXiv:2403.13787 (2024).
>
> [2] Cui, Ganqu, et al. "Ultrafeedback: Boosting language models with high-quality feedback." arXiv preprint arXiv:2310.01377 (2023).
>
> [3] Wang, Zhilin, et al. "HelpSteer2: Open-source dataset for training top-performing reward models." arXiv preprint arXiv:2406.08673 (2024).
>
> [4] Bai, Yuntao, et al. "Training a helpful and harmless assistant with reinforcement learning from human feedback." arXiv preprint arXiv:2204.05862 (2022).
>
> [5] Ganguli, Deep, et al. "Red teaming language models to reduce harms: Methods, scaling behaviors, and lessons learned." arXiv preprint arXiv:2209.07858 (2022).
>
> [6] Touvron, Hugo, et al. "Llama 2: Open foundation and fine-tuned chat models." arXiv preprint arXiv:2307.09288 (2023).
>
> [7] Dubois, Yann, et al. "Length-controlled alpacaeval: A simple way to debias automatic evaluators." arXiv preprint arXiv:2404.04475 (2024).
>
> [8] Eisenstein, Jacob, et al. "Helping or herding? reward model ensembles mitigate but do not eliminate reward hacking." arXiv preprint arXiv:2312.09244 (2023).
>
> [9] Wang, Haoxiang, et al. "Interpretable Preferences via Multi-Objective Reward Modeling and Mixture-of-Experts." arXiv preprint arXiv:2406.12845 (2024).
>
> [10] Zhang, Lunjun, et al. "Generative verifiers: Reward modeling as next-token prediction." arXiv preprint arXiv:2408.15240 (2024).
>
> [11] Ankner, Zachary, et al. "Critique-out-loud reward models." arXiv preprint arXiv:2408.11791 (2024).

---

> > ### Comment · Reviewer_N8sV · 2024-11-21
> > **Thanks for the detailed response**
> >
> > Thanks to the authors' response and additional experiments, my question was well answered. I carefully read the response and other reviewers' comments, and I decided to raise the score to 6 points to agree to accept this paper. I hope the author can make revisions based on other reviewers' comments.

---

> > > ### Author Response · Authors · 2024-11-22
> > > **Thanks for the valuable feedback**
> > >
> > > Thank you for your kind words and for taking the time to review our detailed response. We are pleased that your concerns have been addressed. Thank you again for the time and effort you have dedicated to reviewing our paper and providing valuable feedback. We sincerely appreciate your recognition of our work and will carefully revise the paper based on your and other reviewers’ comments.

---

### Meta-Review · Area_Chair_867Z · 2024-12-05

**Metareview:**

**(a) Summarize the scientific claims and findings**

The paper introduces RMB (Reward Model Benchmark), a comprehensive benchmark designed to evaluate reward models (RMs) used in aligning large language models (LLMs). RMB includes 49 fine-grained real-world scenarios with two alignment categories: helpfulness and harmlessness. The benchmark employs pairwise and Best-of-N (BoN) evaluation methods, offering nuanced insights into RM performance. Experimental results show RMB’s strong correlation with downstream task performance, and the paper identifies key challenges, such as the trade-off between helpfulness and harmlessness. Additionally, it highlights the promise of generative RMs over traditional discriminative approaches.

**(b) Strengths of the paper**

1. **Novelty and Scope**: RMB addresses gaps in existing benchmarks, offering broader task coverage, advanced evaluation paradigms, and nuanced harmlessness testing.
2. **Comprehensive Evaluation**: It uses both pairwise and BoN evaluations, with BoN correlating better with downstream alignment tasks.
3. **Insightful Analysis**: The paper explores trade-offs between alignment objectives and evaluates generative RMs, providing actionable insights for future work.
4. **Strong Empirical Evidence**: Extensive experiments validate RMB’s effectiveness and its correlation with downstream alignment task performance.
5. **Clear Presentation**: The paper is well-written, with a clear structure and detailed explanations.

**(c) Weaknesses of the paper**

1. **Human Annotation Sample Size**: The reliance on 200 human-annotated samples is a limitation, though mitigated by validation and iterative improvements using a golden set.
2. **Potential Bias from GPT-4 Scoring**: While the authors provided evidence of alignment with human judgments and validated their approach with alternative methods, reliance on a single model remains a potential issue.
3. **Limited RLHF Experiments**: The paper focuses on BoN rather than full reinforcement learning fine-tuning (RLHF) due to resource constraints, which may limit its generalization to other RM applications.
4. **Circular Dependency**: RMB relies on responses from current LLMs, which could create challenges in evaluating reward models for future LLM capabilities.

**(d) Provide the most important reasons for the decision.**

The paper addresses an increasingly important research problem in the community: it provides a significant contribution to the evaluation of reward models in LLM alignment through a novel benchmark and rigorous empirical analysis. Despite some limitations, the strengths, particularly in the importance of the problem and the shared insights, justify its acceptance.

**Additional Comments On Reviewer Discussion:**

The reviewers provided thoughtful feedback, which the authors addressed:

1. **Comparison with Existing Benchmarks**: The authors clarified RMB’s advantages over previous benchmarks, adding detailed comparisons and revising the manuscript accordingly.
2. **Trade-offs Between Helpfulness and Harmlessness**: The authors provided deeper insights into the intrinsic conflicts and stylistic differences between the two objectives, supported by theoretical and empirical evidence.
3. **Validation of Annotations**: The authors demonstrated high agreement between GPT-4 annotations, human evaluations, and alternative models, mitigating concerns about bias.
4. **Limitations in RLHF Experiments**: While full RLHF experiments remain a future direction, the authors conducted additional alignment experiments during the rebuttal phase, strengthening their claims.

These discussions and updates improved the paper's clarity and robustness.

---

### Decision · Program_Chairs · 2025-01-22

Accept (Poster)